# Advancing health equity in cancer care: The lived experiences of poverty and access to lung cancer screening

Ambreen Sayani[1,2]*, Mandana Vahabi[3,4], Mary Ann O'Brien[1,5], Geoffrey Liu[6,7], Stephen Hwang[2,8], Peter Selby[5,6,9,10], Erika Nicholson[11], Meredith Giuliani[7], Lawson Eng[7], Aisha Lofters[1,2,4,5,6,12]

1 Women's College Research Institute, Women's College Hospital, Toronto, Ontario, Canada, 2 MAP Centre for Urban Health Solutions, Li Ka Shing Knowledge Institute, St. Michael's Hospital, Toronto, Ontario, Canada, 3 Daphne Cockwell School of Nursing, Ryerson University, Toronto, Ontario, Canada, 4 ICES, Toronto, Ontario, Canada, 5 Department of Family and Community Medicine, University of Toronto, Toronto, Ontario, Canada, 6 Dalla Lana School of Public Health, Toronto, Ontario, Canada, 7 Princess Margaret Cancer Centre, Toronto, Ontario, Canada, 8 Department of Medicine, University of Toronto, Toronto, Ontario, Canada, 9 Campbell Family Research Institute, Centre for Addiction and Mental Health, Toronto, Ontario, Canada, 10 Department of Psychiatry, University of Toronto, Toronto, Ontario, Canada, 11 Canadian Partnership Against Cancer, Toronto, Ontario, Canada, 12 Department of Family Medicine, Women's College Hospital, Toronto, Ontario, Canada

* ambreen.sayani@wchospital.ca

**Data Availability Statement:** All relevant data are within the manuscript and its Supporting Information files.

## Abstract

### Background

Individuals living with low income are more likely to smoke, have a higher risk of lung cancer, and are less likely to participate in preventative healthcare (i.e., low-dose computed tomography (LDCT) for lung cancer screening), leading to equity concerns. To inform the delivery of an organized pilot lung cancer screening program in Ontario, we sought to contextualize the lived experiences of poverty and the choice to participate in lung cancer screening.

### Methods

At three Toronto academic primary-care clinics, high-risk screen-eligible patients who chose or declined LDCT screening were consented; sociodemographic data was collected. Qualitative interviews were conducted. Theoretical thematic analysis was used to organize, describe and interpret the data using the morphogenetic approach as a guiding theoretical lens.

### Results

Eight participants chose to undergo screening; ten did not. From interviews, we identified three themes: Pathways of disadvantage (social trajectories of events that influence lung-cancer risk and health-seeking behaviour), lung-cancer risk and early detection (upstream factors that shape smoking behaviour and lung-cancer screening choices), and safe spaces of care (care that is free of bias, conflict, criticism, or potentially threatening actions, ideas or conversations). We illuminate how 'choice' is contextual to the availability of material

**Funding:** This study was funded through grant # 705592 from the Canadian Institutes of Health Research (https://cihr-irsc.gc.ca/e/193.html) and Canadian Cancer Society (https://www.cancer.ca/) awarded to AL and GL. The funding bodies had no role in the design of the study, data collection, analysis, interpretation of data and writing of the manuscript.

**Competing interests:** AS is supported by a Postdoctoral Fellowship Award from the Canadian Institutes for Health Research. AL is supported by a New Investigator Award from the Canadian Institutes for Health Research, as Clinician Scientist by the Department of Family Medicine at the University of Toronto and as Chair of Implementation Science at the Peter Gilgan Centre for Women's Cancers at Women's College Hospital in partnership with the Canadian Cancer Society. AL is the Provincial Primary Care Lead for Cancer Screening at Ontario Health (Cancer Care Ontario). GL is supported by the Lusi Wong Early Detection of Lung Cancer Program and the Alan B. Brown Chair. GL has received funding from an unrestricted Boehringer Ingerheim grant to develop and implement electronic methods of identifying patients suitable for lung cancer screening in family physician offices. PS is a Clinician Scientist funded by Centre of Addiction and Mental Health, Department of Family and Community Medicine and the Ministry of Health and Long Term Care to direct the STOP program- An Ontario wide smoking cessation program focused on reducing inequities of access for discriminated and underrepresented populations. He also receives support from the Medical Psychiatry Alliance. He has independent grants from the GRAND program, created as an arms-length peer reviewed program for smoking cessation. He is an advisor to both Cancer Care Ontario and Canadian Partnership Against Cancer (CPAC) on tobacco cessation in patients with Cancer. EN is an employee of the CPAC, a pan-Canadian health organization, funded by Health Canada. This does not alter our adherence to PLOS ONE policies on sharing data and materials.

resources such as income and housing, and how 'choice' is influenced by having access to spaces of care that are free of judgement and personal bias.

## Conclusion

Underserved populations will require multiprong interventions that work at the individual, system and structural level to reduce inequities in lung-cancer risk and access to healthcare services such as cancer screening.

## Introduction

Lung cancer is one of the most commonly diagnosed cancers in Canada, and is responsible for a quarter of all cancer-related deaths [1]. Screening using low-dose computed tomography (LDCT) has the potential to diagnose lung cancer at an earlier stage thereby increasing the likelihood of curative therapy [2]. Effective screening interventions must reach populations at high-risk. Thus, engagement with individuals between the ages of 55–74 years, who have smoked cigarettes daily for a minimum of 20 years is critical for success [3].

Historically, individuals with lower levels of income and/or education, or recent immigrants are least likely to undertake preventative health practices such as cancer screening [4]. For lung cancer screening this is particularly problematic since the greatest lung cancer incidence is found in populations disadvantaged by a variety of social determinants such as income and education [5]. Lung cancer screening therefore presents two very specific challenges: (i) a disease distribution that is proportionate to the degree of social disadvantage; and (ii) an opportunity to screen for the disease in the target population, but with a potentially low uptake.

Quantitative and mixed methods studies document that individuals at high-risk of developing lung cancer who decline lung cancer screening are more likely to be active smokers, report personal blame or stigma, perceive lower benefit from undertaking the test, or have fatalistic beliefs about their own well-being [6–8]. Current recommendations to target this group therefore are aimed at enhancing invitation strategies [8], developing interventions to support decision-making and decreasing knowledge conflicts about screening between physicians and patients [7]. In this context, a major knowledge gap is an understanding of the lived experiences of individuals who are socially disadvantaged and the factors that influence their choice to access preventative health services, such as LDCT lung cancer screening. Thus, to address this knowledge gap, we conducted a qualitative study in Ontario where the implementation of lung cancer screening is currently being piloted.

## Methods

### Study design

Theoretical thematic analysis (TTA) is a qualitative research methodology that provides a systematic way to organize, describe and interpret data using a guiding theoretical lens [9]. TTA can be used to socially contextualize data [9] and inform the development of evidence-based policies applicable to the real-world setting of patients, providers and policy makers [10]. TTA is useful in analyzing participant perspectives, particularly if the perspectives are different, by highlighting similarities and differences across datasets to generate unanticipated insights [9, 11]. We therefore used TTA to answer the following research question: What are the perspectives of individuals living with low income towards lung cancer screening and the choice to access or not access screening?

## Theoretical framework

The morphogenetic approach is a critical realist theory which posits that three primary causal powers determine individual behaviour; namely, structural relationships in society, sociocultural interactions between individuals, and individual agency [12, 13]. The term "morpho" takes its roots from the word shape, and "genesis" from the word shaping [12]. To be morphogenetic therefore implies taking action for change and the processes that elaborate or change system and/or structure. On the contrary, to be morphostatic implies to resist change and to maintain the status quo [12].

The morphogenetic approach has been described as the preferential framework with which to understand agency in higher education [14, 15] and individual transformation in criminology [16]. In this study we introduce the morphogenetic approach to health services research and describe how it can be used to critically contextualize the enablers of health seeking behaviour in health promotion practice.

The core assumptions of the morphogenetic approach (Fig 1) are: (i) that structural relationships pre-exist in society (T1); (ii) that these social structures influence social interaction (T2); (iii) that causal relationships exist between individuals and groups that influence social interaction (T3); and (iv) that social interaction can influence individual agency and social structure (T4) [12]. Subsequently, we applied these core assumptions to theorize the 'choice' to access lung cancer screening for individuals living with low income in the following way (Fig 1) [17]:

(1) T1: Social, political, cultural and economic contexts influence current lung cancer risk;

(2) T2: Socially-disadvantaged patients interact with the health system through various contexts;

(3) T3: These interactions condition patients to respond in a certain way;

(4) T4: The choice to seek preventative health care is based on these interactions and can lead to:

 a. T4a: Morphostasis or structural reproduction, by negative perspectives towards health and health seeking choices, that subsequently maintains the status quo and perpetuates lung cancer risk; or

 b. T4b: Morphogenesis or structural elaboration, by positive perspectives towards health and health seeking choices that subsequently transforms behaviour and potentially lowers lung cancer risk.

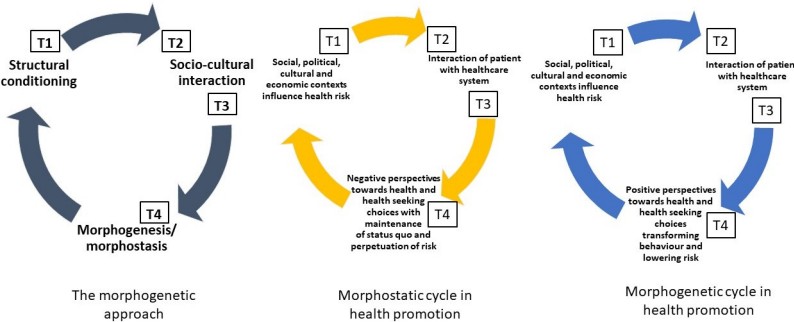

**Fig 1. Theorizing choice in health promotion though the morphogenetic approach.**

## Participant recruitment

Participants were recruited from three primary care sites in downtown Toronto that serve a wide demographic of patients, including those experiencing low income, homelessness and addictions. Eligible patients met the provincial referral criteria for individuals at high-risk for developing lung cancer: i) were between 55 to 74 years of age, ii) had smoked daily for at least 20 years, and iii) did not have any active cancers. We followed provincial exclusion criteria: prior lung cancer, under surveillance for lung nodules, or history of hemoptysis, unexplained weight loss or a life-threatening condition. No prior association existed between the principal researcher and participants prior to the commencement of the study.

Participants were recruited from March 2019 to January 2020 and were categorised as screeners or non-screeners in the following way:

**Screeners.** Screeners were study-eligible participants, referred by their healthcare provider, who chose to screen with LDCT. Upon receiving a referral, the patient navigator at the provincial lung cancer pilot screening site called patients to determine eligibility by conducting a risk assessment using PLCO$_{M2012}$ criteria [18]. Patients were study-eligible if their calculated risk score was >2% in the next six years [19], had recently received a LDCT, and consented to be contacted for a research interview. The 2% threshold is the current screening eligibility criterion being used in the province of Ontario for lung cancer screening [20].

**Non-screeners.** Non-screeners were patients who chose not to undergo LDCT lung cancer screening. Recruitment of non-screeners through the clinical sites was challenging as eligible participants who were not interested in screening were also not interested in participating in the research study. We therefore used a technique called derived rapport [21] in order to recruit non-screeners by leveraging one health provider's established relationship with a low-income patient community. Thus, patients were assessed for lung cancer screening eligibility by that health provider. When participants were not interested in screening they were informed about the study, and introduced to the first author (AS) who was on-site to conduct interviews immediately thereafter.

## Data collection

Interview guides were developed using the guiding theoretical lens. Separate interview guides were developed for screeners (S1 File) and non-screeners (S2 File) in order to explore different choices in health seeking behaviour. The interview guides included questions on smoking history, smoking cessation, living conditions, and perspectives on health and preventative health care such as lung cancer screening. Semi-structured interviews were conducted with open-ended questions to encourage a participant-led dialogue.

All interviews were conducted by the first author (AS). The Research Ethics Board at Unity Health Toronto provided ethics approval for the study. This included the board's approval to obtain verbal consent from all participants before beginning the research interview. Therefore, prior to the interview all participants were given information about the purpose of the research study and expectations from the interview including interview duration and usage of data. The first author (AS) discussed the ways in which results from the research study would be disseminated and sought approval to record participants verbal consent. After participants agreed to proceed, audio recording was initiated, verbal consent taken, and included in transcribed files as part of data collection. This process was approved by the REB at Unity Health Toronto.

Non-screeners were interviewed face-to-face in the clinic space of the healthcare provider who had facilitated their recruitment immediately after they had provided consent. Screeners were interviewed four to six weeks after they underwent LDCT lung cancer screening. Interviews with screeners were performed via telephone in order to reduce attrition as face-to-face

interview appointments in low-income populations can be hindered by competing priorities and constrained resources that detract from research participation [21]. Interviews lasted approximately 40 to 60 minutes.

All interviews were audio-recorded then transcribed by a professional transcriptionist. Audio files were sent to the transcriptionist on the day of the interviews and transcripts were reread (by AS) for accuracy within two days. Personal identification was removed from the audio and transcribed files, and pseudonyms were given to all participants. Field notes taken during the interviews were added to the study files. Interviewing continued till the point of conceptual saturation.

Participants provided sociodemographic data using a pre-existing health equity survey [22] to capture data including information on the participants' housing status, ethnicity, language, sexual orientation, disability status, and household income.

## Data analysis

We used TTA to organize, describe and interpret the data [9]. The first (AS) and last author (AL) read through the data multiple times to search for repeated patterns of meaning [9] and to develop a coding framework. NVivo 12 software was used to organize the data manually and to code units with similar meaning, even if they did not sync with the theoretical inquiry. This was important to ensure that patterns in text were not missed or disregarded [23]. Line-by-line coding was applied to the textual data (AS), and codes were developed iteratively during weekly consultations between two authors (AS, AL). AS is a medical doctor and critical qualitative researcher. AL is a family physician and clinical epidemiologist. Both AS and AL research health inequities. The larger interdisciplinary research team (all authors) met regularly during all research phases to discuss challenges and emerging themes, and provided an opportunity to regularly peer debrief until we reached conceptual saturation [24] and our final themes. The extended research team comprised of clinical oncologists (GL, LE), radiation oncologists (MG), clinician scientists in addiction and mental health (PS) and homelessness (SH) as well a federal level policy maker that is involved in the implementation of lung cancer screening in Canada (EN). As a team we were concerned with themes occurring at the latent level (Braun & Clarke, 2006). Latent themes seek to explain or understand why participants said or acted in a certain way, and as such differ from semantic themes which are limited to a description of data rather than its interpretation [25].

## Results

### Characteristics of study sample

We interviewed 18 screen-eligible individuals (eight screeners, ten non-screeners), who were primarily male, white and Canadian-born (Table 1). At the time of interview, seven participants were living in a homeless shelter in downtown Toronto. Eight participants lived in subsidized apartments through Toronto Community Housing, and the remaining three were renting their accommodations. Most participants reported annual incomes below $30,000 a year, while two reported income levels between $30,000 and $59,000. Participants who declined to undergo lung cancer screening were more likely to be living in the homeless shelter (seven participants) and report incomes below $30,000 (nine participants).

### Themes

Fig 2, shows a thematic map connecting our three main themes: Pathways of disadvantage, upstream determinants of lung cancer risk and early detection, and safe spaces of care.

**Table 1. Participant sociodemographic information.**

|  | Screeners | Non-Screeners |
|---|---|---|
| Total | 8 | 10 |
| **Age categories** | | |
| 50–59 | 1 | 2 |
| 60–69 | 3 | 7 |
| 70+ years | 4 | 1 |
| **Sex** | | |
| Male | 7 | 10 |
| Female | 1 | 0 |
| **Immigrant status** | | |
| Foreign-born | 2 | 4 |
| Canadian-born | 6 | 6 |
| **Ethnicity** | | |
| White–(North American, European) | 6 | 7 |
| Asian—East (e.g., Chinese, Japanese, Korean)/ South (e.g., Indian, Pakistani, Sri Lankan) | 1 | 1 |
| Other (including Indigenous, Indian and Latin American | 1 | 2 |
| **Self reported income (in Canadian dollars)** | | |
| 0–29,999 | 6 | 9 |
| 30,000–59,000 | 2 | 0 |
| 60,000–89,000 | 0 | 0 |
| 90,000–119,000 | 0 | 0 |
| 120,000 plus | 0 | 0 |
| Missing | 0 | 1 |
| **Sexual orientation** | | |
| Heterosexual | 7 | 10 |
| Gay/Bisexual/Lesbian/Queer/Two-Spirit/Other | 1 | 0 |
| **Housing** | | |
| Own Home | 0 | 0 |
| Renting Home | 3 | 0 |
| Community housing | 5 | 3 |
| Homeless shelter | 0 | 7 |

An overview of the themes is discussed below. The complete set of themes, subthemes and illustrative quotes that were derived from the transcripts is shown in Table 2.

**Pathways of disadvantage.** Clustering of disadvantage is a phenomenon whereby people who are disadvantaged in one area, such as income, are also disadvantaged in other areas, such as housing, food security, and educational opportunities [26]. Graham et al [27] have described the clustering of disadvantage from a life course perspective in relation to smoking behaviour to illuminate how poor circumstances in childhood influence future educational achievement and economic prospects contributing to a pathway of disadvantage that is deeply entwined with adult smoking behaviour. Given that it was neither the purpose nor scope of this study to identify the links between childhood circumstance and adult lung cancer risk and health seeking choices we bracketed our analysis of the life course to begin with working conditions and to end with current risk for lung cancer. The theme pathways of disadvantage, therefore, describes a social trajectory of events over the life course that influence lung cancer risk and health seeking behaviour.

In our study, all participants described precarious working conditions leading to economic instability and fluidity of material resources such as income. Participants also recounted

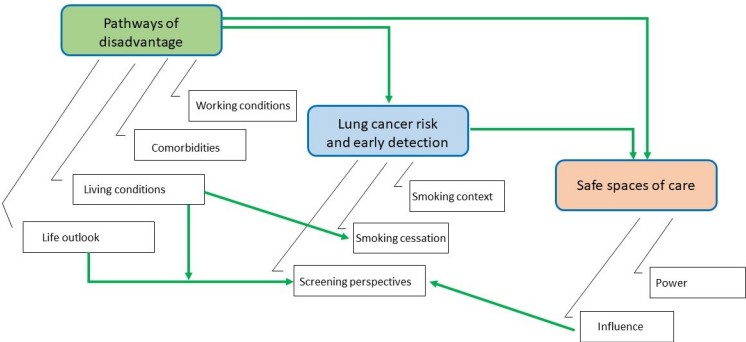

**Fig 2. Thematic map.** Understanding the perspectives on choice for access to lung cancer screening for individuals living with low income.

physical or mental disabilities related to their working conditions or life circumstance. Specifically, half of the participants reported severely restricted current mobility due to spinal/joint injuries and arthritis that were associated with prior employment in manual labour. All participants had experience of being de-housed. Participants recounted the reasons for experiencing housing insecurity as income precarity and associated comorbidities. For participants who chose to undergo lung cancer screening, housing was described as a key determinant that influenced positive life outlook and perspectives towards preventative health care in general.

**Upstream determinants of lung cancer risk and early detection.** Smoking is the single largest risk factor for lung cancer and an individual's smoking history is a key enrolment criterion for the lung cancer screening pilot in Ontario. Accordingly, we define the theme upstream determinants of lung cancer risk and early detection as the social and economic factors which shape smoking behaviour and lung cancer screening choices.

All participants described the context in which they initiated smoking at an early age, recounting in detail the social acceptability of smoking, availability of cigarettes at an affordable price point, a lack of awareness of the associated harms, and mass media advertising which promoted its use. All participants recounted in detail how they had attempted smoking cessation. At the time of interviews however, only three participants had successfully been able to quit smoking and an additional four were actively enrolled in a smoking cessation program. Of the three participants able to quit smoking, one was living in subsidized housing and the other two were renting their homes (reporting incomes between $30–59,000 a year). These participants described how having access to material resources (such as income and housing) directly influenced their ability to choose and benefit from smoking cessation.

The three participants who had successfully quit smoking chose to be screened for lung cancer. Access to material resources had a direct influence on lung cancer screening perspectives. All eight participants who had undergone screening had their own place to stay (subsidized housing or rent). Five of eight reported having participated in other preventative health checks such as colon cancer screening. Participants who chose not to screen had competing priorities such as making ends meet, unemployment and housing security that were more pressing needs than screening for an illness which could possibly occur several years later.

**Safe spaces of care.** A safe space of care is a place that is free of bias, judgment, conflict, criticism, or potentially threatening actions, ideas or conversations. For our study, we limited this theme to the lived experiences of participants' encounters with front line health workers and the impact of these experiences on their subsequent health promoting choices. We build on the definitions of power (the ability of an agent to act), 'power over' (a social relation in

**Table 2. Themes, subthemes, and illustrative quotes from interview transcripts.**

| THEME—SUBTHEME | ILLUSTRATIVE QUOTES |
|---|---|
| **Pathways of disadvantage: a social trajectory over the life course that significantly influences health risk and health promotion practices** | |
| **1. WORKING CONDITIONS** | "I'm always worried about tomorrow. It's like okay, I have some (money) this week okay, let's see if I can get some more for next week" (Jack).<br>"When you drive a taxi especially. . . . . .. your income is very fluid. You can do well one day and just barely cover your costs the next. So, yeah, there are times when it has been difficult to make ends meet" (Caitlyn).<br>"I was working with like a subcontractor. . . I was paid by the job, yeah. I was paid by the job and not by hour, not by hour. . . I used to work and then I got sick and when I get in and out from the hospital, I lost my job and yeah I lost my job, I don't have no income" (Rocco.) |
| **2. COMORBIDITIES** | "My body. It's like the problems with my back and everything just didn't instantly happen. It's, it was going on with me for like years you know and even when I was 40 years old, 45 years old like my back just pounded like crazy like I was always in a lot of pain and I ran you know dozers and scrapers for years" (Philip).<br>"You know it does not immediately appear, like appearance, like just stress. But after a while you feel like tired, headache, irritate, you know the mood is sometimes just uncontrollable something like that. You want to escape. You want to be alone. These kind of things (Walter)". |
| **3. LIVING CONDITIONS** | "Well I pretty much wound up in shelters because like when you run heavy equipment it's not like I was working all year round. I would only work maybe eight months out of each year you know and it was all union work so sometimes the jobs the union would send me to would be six months, some would be a year, some would be longer so I was in and out of shelters you know "(Philip).<br>"Oh, the housing I mean it's $2,000.00 a month for an apartment and you got to have first and last. Where is a guy who's working, or in a hostel, (or) you know even if he gets a decent job that's paying $20.00 an hour, he's going to have to stay there for three months to make up enough money and that's if everything goes perfect for him at $20.00 an hour. I mean you'll have enough to, because he might get a place for $1,700.00 you never know but I mean then you might have to travel right across town to get to his job too, so that's another $150.00 that he's going to have to spend a month" (Barry). |
| **4. LIFE OUTLOOK** | "I just turned 60, yeah. Monday, I turned 60. Well I'm over the hill. Yeah, I haven't got, and what I got, a couple of years, a couple of years left? I think so. Something like that" (Jarod).<br>"For me it was because I had housing and I wanted to look into my health. But for guys on the street and in hostels man, they have other priorities. They need housing first before they can do anything. And that's going to be their problem and like yeah, I'll do that, sure but right now I'm doing this, I need housing you know; you're way down at the bottom of their priority list right now because for me anyway, you would be too because I need housing. I got to get myself quiet and healthy first and a sane mind at least to be able to make an appointment but I mean you're not going to do that when you're fighting with residents and staff at a shelter or on the streets" (Barry). |
| **Upstream determinants of Lung cancer risk and early detection: the social and economic factors which shape smoking behaviour and lung cancer screening choices** | |
| **1. SMOKING CONTEXT** | "We used to smoke behind the yard you know in the school during school break you know" (Harold).<br>"I guess like 14, 15; 15 years of age and I guess I was starting. Other people were starting to smoke so I thought yeah, I'll go along with it, with the crew or something like that. . . Cigarettes in those days were $0.45 cents I think for a package of cigarettes so even with my measly earnings delivering newspapers I was able to purchase cigarettes so the price was right and then they were available and there was no advertisements about harm to your health on at that time. There was very little said about cigarette smoking, and it was heavily propagandized in Hollywood films and that, so you know (it was) just a thing of the time" (Patrick).<br>"Things were very cheap (where I was) . . . I paid $0.25 for a large pack of cigarettes so it was easy to be . . .a heavy chain smoker" (Barry). |
| **2. SMOKING CESSATION** | "I quit smoking for over a year but then I moved into the building where I am now at Sherbourne and Dundas and the stress of all the stuff going on in the building and all the crackheads and drug dealers on the corner I started smoking again" (Jaden).<br>"Probably seeing other people smoke, smelling it, you know like the smoke in the air" (Kane).<br>"Yeah, I tried before but like the stress doesn't help much but right now my stress level is pretty good. Like some days I don't even smoke a cigarette. Some days I smoke one, two like for the last three months and it's like okay, if I have no stress and I'm happy and walking around and I can go like two, three days without having a cigarette" (Jack). |
| **3. SCREENING PERSPECTIVES** | "Everybody has priorities. You have priorities in life. You have choices to make every day. You have a priority whether you want to do something next week or not or whether you want to make a plan . . . Sometimes you just put it aside until you need to use it and so people that are homeless okay, they're not looking to get help at that end. They're looking to get help to find a home . . . Well (smoking cessation for me) was planned because when I was smoking a pack and a half a day and then I cut it down to a package and a quarter a day, then I cut it down to a pack a day, then down to ¾ of a pack a day, then I cut it down to a half a pack a day and then I cut it down to six cigarettes a day. At that point I went to my health provider and got on a plan to get the patch for I think it was two weeks on my arm and to cut out smoking completely" (Hank).<br>"When you're worried about issues of making ends meet, there are several things that could happen. When you're unemployed you can very easily seep into apathy and when you're apathetic you don't get up off your butt and go and have a test to see if you've got lung cancer. I think that's probably, you know that's my wisest words on it. I think you know when you're, when everything is okay and especially if you're retired it's much easier for people to go to a clinic and have a test which only takes what ten minutes? But when you're unemployed and you know you've got economic problems then you might think to yourself well you know I'm worried about getting a job and keeping my house and so forth and you think well I'll put that off. I've got other things to worry about right now you know' (Harold). |

*(Continued)*

**Table 2.** (Continued)

| THEME—SUBTHEME | ILLUSTRATIVE QUOTES |
|---|---|
| **Safe spaces of care: a place of clinical care which is free of bias, judgement, conflict, criticism, or potentially threatening actions, ideas or conversations** | |
| **1. POWER** | "The most important impact is not having less money to spend, it is that you have to face the government workers. They are not very sympathetic to that thing. They treat you like you are a beggar or pest or something like that. It's that kind of feeling. It makes you feel that you don't want to come to them to ask for help" (Walter). |
| | "They helped me with 60% of the cost of my prescription. One day she said to me that I am taking advantage of the system and I don't know how? How with all my gratitude and the love that I have for her? It made me feel like shit. It made me feel like when I was a little kid and my mother put me on the street to beg for a piece of bread, looking for something to eat in the garbage cans, and a few tomatoes" (Rocco). |
| **2. INFLUENCE** | "I got treated really bad from doctors and I didn't want to go into another office and get treated like a piece of garbage because you come out and you feel twice as bad" (Damien). |
| | "Well she has a street-like attitude about things . . . she doesn't have to talk with fancy words, and you know. We get along well like our rapport is quite good. She knows where I'm from and you know a lot more than most people know about me so, yeah, she's somebody I feel I can trust you know. You can tell her things and it's; she understands what you're talking about and stuff like that . . . You're not actually on a visit there, you're just talking to her and its straight talk, like she doesn't treat you like you're below her or anything like that" (Jaden). |
| | "The more you trust, the more you trust that person, the more you're going to be willing when they do suggest things" (Caitlyn). |

which one person/group dominates to shape the available 'choices' of a subjugated person/group) [28] and influence (the power to have a direct effect on someone so as to change their actions in an important way) to contextualise the health seeking choices of participants ~ i.e. morphogenesis (accessing lung cancer screening) or morphostasis (declining lung cancer screening).

An important observation in our study was that all three participants who were renting their home chose to be screened for lung cancer, whilst all seven participants in the homeless shelter declined. Of eight participants who were living in subsidized housing at the time of the interviews, five chose to be screened for lung cancer, and three declined. All five who chose to be screened reported having a collaborative/supportive/non-judgemental relationship with their providers that promoted their personal health and wellbeing. These participants shared experiences of care that were sensitive to their life circumstances. Subsequently, participants recounted their clinical encounters as pleasant and comfortable. In contrast to these experiences, the three participants living in subsidized housing who declined lung cancer screening described encounters that were judgmental and insensitive to the sociohistorical realities that constituted their lives. Participants used words such as, "worthlessness", "garbage", "beggar" and "pest", as they recounted these experiences.

## Discussion

Despite a publicly funded health system in Canada, participation in population-wide screening programs has not been universal. Differences in screening uptake are associated with income, education and immigrant status [4]. Individuals who live with greater degrees of social disadvantage have a higher risk of some cancers and poorer overall survival; this is directly correlated with the social determinants of health and how they intersect across the cancer care continuum [29]. Therefore, new interventions such as lung cancer screening must take into consideration differences in utilization and needs, which are based on social location. This is key to preventing an inadvertent widening of the health equity gap that already exists between population groups. Our study sought to fill this knowledge gap in the context of lung cancer screening which is currently being pilot tested in the province of Ontario. Specifically, we wanted to understand the perspectives on choice for lung cancer screening in patients living with low income in order to inform the design and delivery of lung cancer screening as an organized program.

Through semi-structured interviews with individuals living with low income who chose to undergo or not undergo lung cancer screening, we found that participants' interest in screening depended on the availability of adequate housing, which subsequently empowered participants to seek care to advance their health and wellbeing. Further, clinical encounters created a space that needed to be navigated without personal bias and with sociohistorical sensitivity. Participants who recounted their clinical encounters as places of judgment-free care had meaningful relationships with their physicians, and they subsequently trusted their physicians' judgement and recommendations.

All participants in our study initiated smoking at a very early age, recounting cigarettes as being available, accessible and, "the thing to do". Much of this context is related to the mass-marketing campaigns that ran across media stations and sensationalized smoking in the 1950's and 1960's [30]. Since then, policies enacted to limit the marketing and use of tobacco products in public spaces have led to decreases in smoking rates across Canada. However, this distribution is skewed such that individuals living with socioeconomic disadvantage are more likely to smoke and less likely to cease smoking [31].

Recent studies have highlighted that willingness to quit smoking is equal across social classes; however, socially-disadvantaged smokers are more likely to: (i) live in an environment where they socialize and work with other socially-disadvantaged people for whom smoking is considered acceptable [32]; (ii) use smoking as a way to relax and cope with high daily stress levels [33]; and (iii) report experiences of disadvantaged childhood, educational and employment trajectories that shape the pattern and frequency of cigarette consumption [27]. These findings are closely reflected in the lived experiences of our study participants all of whom had attempted smoking cessation at least once, yet only three had quit successfully.

In our study, the three participants who had successfully quit smoking reportedly quit because they became more proactive towards their own health once they had found housing and all three took part in lung cancer screening. Participants identified living conditions (housing) to be a key determinant of health and wellbeing, particularly in the context of a preventative health check such as lung cancer screening. Significantly, only those participants who had their own place to call home felt empowered enough to seek opportunities to advance their health and then take action to utilize the service of screening. Participants who chose not to undergo screening lived in diverse housing situations that were generally less secure (subsidized housing; homeless shelter). For these participants, lack of adequate housing was one barrier to care.

Another barrier was the attitude and influence of treating physicians, which determined participants' willingness to continue to engage with the health system. In Canada, physicians are gatekeepers to and proponents of lung cancer screening and therefore have a critical role in ensuring equitable healthcare delivery. We have described physician perspectives on access to lung cancer screening for individuals living with low income elsewhere; a key finding was that an equity-oriented approach and attention to the upstream determinants of lung cancer by physicians will be needed in order to improve equitable access to lung cancer screening [34]. Our findings suggest that physicians and other healthcare practitioners must learn how to deliver care that is free of personal biases to prevent the perpetuation of oppression and the systemic reproduction of health inequities.

According to the "inverse equity hypothesis" [35] population-based health interventions are more rapidly adopted by the wealthy, a term described as "top inequality" [36]. This is in contrast to "bottom inequality" which refers to the lag in adoption experienced by the poorest when the intervention has reached high coverage across the remainder of the population (Victora et al., 2005). Thus, individuals developing health interventions must recognize patterns of health inequality and adapt policies and processes to mitigate these effects [37] and prevent a widening of the health equity gap between socially advantaged and disadvantaged population groups [38].

Midstream interventions [39, 40] geared at reducing cigarette consumption in the socially-disadvantaged are only likely to be truly successful if incorporated into a broader program that addresses the social context of smoking behaviour. As a standalone intervention, nicotine replacement therapy is unlikely to be highly effective in influencing smoking behaviour and reducing lung cancer risk for those who are socially-disadvantaged. Similarly, lung cancer screening as a midstream public health program may inadvertently increase health inequities unless proactive and multipronged strategies are in place to increase uptake. Screening programs may also neglect the underlying social context of smoking behaviour, such that even if individuals are screened for lung cancer, they may continue to face increased lung cancer risk due to the daily stressors that influence their choice to continue smoking. This approach to care is ignorant of the lived realities of those at high-risk of developing lung cancer, the target population of the lung cancer screening program. An equity-focused approach to screening programs therefore, would demand attention to the social context of cancer risk and utilization rates based on social location.

Health systems can determine how to respond best to inequitable health intervention uptake by engaging with patients to identify strategies that enhance intervention effectiveness [41]. Conducting future patient-oriented research that incorporates the lived experience of those at high-risk for lung cancer to guide the development of a multipronged approach (such as education, resources, and tools) can maximize the availability and accessibility of the intervention to the target population [42]. This approach, called targeting within universalism implies that a universally available intervention must be accompanied by specific uptake strategies that positively discriminate towards those at highest risk [43].

## Strengths and limitations

The focus of our study was on perspectives towards preventative healthcare, particularly lung cancer screening for individuals living with low income. As such, we have offered an analysis of the lived experiences of choice in relation to income and social class, using a critical realist lens termed the morphogenetic approach. We realize that using a different analytical lens to frame our research, such as intersectionality, would have offered alternative themes and policy recommendations; this can be a future research area as interventions are evaluated for their population effectiveness based on differences in social identity. In particular, we recommend future research with racial and ethnic minority groups in order to understand how stigma and/or racism may create further barriers to healthcare. The homogenous nature of our non-screening participant sample reflects how we recruited through the established practice of one health provider. However, by using this approach, our study is able to demonstrate how derived rapport [21] can be used to reach research participants who are both high-risk (in terms of health risk for lung cancer) and hard-to-reach (in terms of difficulty to access for research) for a variety of reasons such as geographic location, economic deprivation, or vulnerability due to stigma and disenfranchisement [44, 45]. As a qualitative research study, we have described our study setting, recruitment strategy and participant population in order to promote transferability of our research findings.

## Conclusion

Inherently, any health promotion program that focuses only on individual uptake, such as choosing to undergo cancer screening, runs the risk of masking the fundamental causes of health conditions and may inadvertently widen health inequities that already exist. Program oversight by a governance structure that is focused on monitoring, tracking and advancing progress towards health equity will be needed. Ultimately, the health system must actively

resist increasing the health equity divide by targeting both upstream social policies (that will ultimately reduce cancer risk), as well as midstream interventions that can enhance health seeking choices. To do this would require an emphasis on the social policies that facilitate stable work, adequate income, and secure housing. One of the ways in which this can be achieved is through the integration of patient-identified priorities. Future research into the lived experiences of patients can continue to provide meaningful insights into what works, and what is needed to enhance health promoting opportunities.

## Supporting information

**S1 File. Interview guide for screeners.**
(PDF)

**S2 File. Interview guide for non-screeners.**
(PDF)

**S3 File. Consolidated criteria for reporting qualitative studies (COREQ) checklist.**
(PDF)

## Acknowledgments

This article includes work from the published dissertation of AS. We would like to thank Dr. Dennis Raphael, Dr. Marina Morrow and Dr. Janet Parsons for their invaluable feedback during the conceptual development of this work.

## Author Contributions

**Conceptualization:** Ambreen Sayani, Mandana Vahabi, Mary Ann O'Brien, Geoffrey Liu, Stephen Hwang, Peter Selby, Erika Nicholson, Meredith Giuliani, Aisha Lofters.

**Data curation:** Ambreen Sayani, Mary Ann O'Brien, Aisha Lofters.

**Formal analysis:** Ambreen Sayani, Mandana Vahabi, Mary Ann O'Brien, Geoffrey Liu, Stephen Hwang, Peter Selby, Aisha Lofters.

**Funding acquisition:** Geoffrey Liu, Aisha Lofters.

**Investigation:** Ambreen Sayani, Mandana Vahabi, Mary Ann O'Brien, Geoffrey Liu, Stephen Hwang, Aisha Lofters.

**Methodology:** Ambreen Sayani, Mandana Vahabi, Mary Ann O'Brien, Geoffrey Liu, Stephen Hwang, Peter Selby, Aisha Lofters.

**Project administration:** Ambreen Sayani, Geoffrey Liu, Aisha Lofters.

**Resources:** Geoffrey Liu, Stephen Hwang, Peter Selby, Aisha Lofters.

**Software:** Aisha Lofters.

**Supervision:** Stephen Hwang, Aisha Lofters.

**Validation:** Ambreen Sayani, Mandana Vahabi, Mary Ann O'Brien, Geoffrey Liu, Stephen Hwang, Peter Selby, Erika Nicholson, Meredith Giuliani, Lawson Eng, Aisha Lofters.

**Visualization:** Ambreen Sayani.

**Writing – original draft:** Ambreen Sayani.

**Writing – review & editing:** Ambreen Sayani, Mandana Vahabi, Mary Ann O'Brien, Geoffrey Liu, Stephen Hwang, Peter Selby, Erika Nicholson, Meredith Giuliani, Lawson Eng, Aisha Lofters.

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
