## [Decision Letter · Decision Letter 0]

9 Mar 2021

PONE-D-21-01483

Advancing Health Equity in Cancer Care: The Lived Experiences of Poverty and Access to Lung Cancer Screening.

PLOS ONE

Dear Dr. Sayani,

Thank you for submitting your manuscript to PLOS ONE. After careful consideration, we feel that it has merit but does not fully meet PLOS ONE’s publication criteria as it currently stands. Specifically, the paper is way too long (~5000 words) and needs to be shorted by half.  The topic is of interest, but one of the reviewers was unclear about the heavy focus on affordable housing as a proxy for SES.  This needs to be explained more clearly.  The limitations of the qualitative approach needs to be mentioned in the discussion and what specific next steps would the author recommend to address obvious disparities in lung cancer screening. However, because the paper addresses a topic of important that has not received much attention in the published literature we invite you to submit a substantial revised and shortened version  of the manuscript that addresses the points raised during the review process.

If you choose to submit a  revised manuscript by Apr 03 2021 11:59PM. If you will need more time than this to complete your revisions, please reply to this message or contact the journal office at plosone@plos.org. Please include the following items when submitting your revised manuscript:

We look forward to receiving your revised manuscript.

Kind regards,

Michael Cummings, PhD

Academic Editor

PLOS ONE

Journal Requirements:

2. In your Methods section, please provide additional information about the participant recruitment method and the demographic details of your participants. Please ensure you have provided sufficient details to replicate the analyses such as:

a) the recruitment date range (month and year),

b) a description of how participants were recruited, and

c) descriptions of where participants were recruited and where the research took place (e.g. name of clincial site).

3. Please state in the manuscript Methods:

- Why written consent could not be obtained

- Whether the Institutional Review Board (IRB) approved use of oral consent

- How oral consent was documented

For more information, please see our guidelines for human subjects research: https://journals.plos.org/plosone/s/submission-guidelines#loc-human-subjects-research

'AS is supported by a Postdoctoral Fellowship Award from the Canadian Institutes for Health Research. AL is supported by a New Investigator Award from the Canadian Institutes for Health Research, as Clinician Scientist by the Department of Family Medicine at the University of Toronto and as Chair of Implementation Science at the Peter Gilgan Centre for Women’s Cancers at Women’s College Hospital in partnership with the Canadian Cancer Society. AL is the Provincial Primary Care Lead for Cancer Screening at Ontario Health (Cancer Care Ontario). GL is supported by the Lusi Wong Early Detection of Lung Cancer Program and the Alan B. Brown Chair. GL has received funding from an unrestricted Boehringer Ingerheim grant to develop and implement electronic methods of identifying patients suitable for lung cancer screening in family physician offices. PS is a Clinician Scientist funded by Centre of Addiction and Mental Health, Department of Family and Community Medicine and the Ministry of Health and Long Term Care to direct the STOP program- An Ontario wide smoking cessation program focused on reducing inequities of access for discriminated and underrepresented populations. He also receives support from the Medical Psychiatry Alliance. He has independent grants from the GRAND program, created as an arms-length peer reviewed program for smoking cessation. He is an advisor to both Cancer Care Ontario and Canadian Partnership Against Cancer (CPAC) on tobacco cessation in patients with Cancer. EN is an employee of the CPAC, a pan-Canadian health organization, funded by Health Canada.'

a. Please confirm that this does not alter your adherence to all PLOS ONE policies on sharing data and materials, by including the following statement: "This does not alter our adherence to  PLOS ONE policies on sharing data and materials.” (as detailed online in our guide for authors http://journals.plos.org/plosone/s/competing-interests).  If there are restrictions on sharing of data and/or materials, please state these.

Please note that we cannot proceed with consideration of your article until this information has been declared.

5. Please ensure that you refer to Figure 2 in your text as, if accepted, production will need this reference to link the reader to the figure.

6. Please include your table as part of your main manuscript and remove the individual file. Please note that supplementary tables should remain as separate "supporting information" files.

7. Please include captions for your Supporting Information files at the end of your manuscript, and update any in-text citations to match accordingly. Please see our Supporting Information guidelines for more information: http://journals.plos.org/plosone/s/supporting-information

8. Your ethics statement should only appear in the Methods section of your manuscript. If your ethics statement is written in any section besides the Methods, please delete it from any other section.

Reviewers' comments:

Reviewer's Responses to Questions

**Comments to the Author**

1. Is the manuscript technically sound, and do the data support the conclusions?

Reviewer #1: Yes

Reviewer #2: Partly

Reviewer #3: Yes

2. Has the statistical analysis been performed appropriately and rigorously? 

Reviewer #1: N/A

Reviewer #2: N/A

Reviewer #3: N/A

3. Have the authors made all data underlying the findings in their manuscript fully available?

Reviewer #1: Yes

Reviewer #2: No

Reviewer #3: Yes

4. Is the manuscript presented in an intelligible fashion and written in standard English?

Reviewer #1: Yes

Reviewer #2: Yes

Reviewer #3: Yes

5. Review Comments to the Author

Reviewer #1: Sayani and colleagues present the results of a qualitative research study aimed at describing barriers and facilitators to lung cancer screening in low income individuals in Toronto. The methodology and theoretical framework are sound, and the findings are important as countries consider how to target those at the highest risk for lung cancer and are also the most disadvantaged due to socioeconomic status. I have the following comments:

1) Overall, the manuscript is quite lengthy of a read at over 5000 words. I understand that PlosOne does not have a limit on words but would encourage the authors to summarize methodology and limitations and strengths sections further.

2) Similarly, the authors chose to embed representative quotes into the body of the text which extends the length of the manuscript. I would suggest considering instead constructing a table that pulls from the themes, subthemes and key findings presented in Figure 2 and putting in representative quotes into the table. If the authors chose to leave the quotes embedded, I would suggest reducing the length of the quotes. For example, participant Philip’s quotes could be shortened considerably. In addition, including quotes from the other 18 participants rather than utilizing the same participants might better support the themes and conclusions identified.

3) In the discussion section, the paragraph that discusses gender-based differences in smoking is not particularly relevant to the focus of the manuscript especially given the low number of female participants and distracts. I would suggest removing.

Reviewer #2: This interesting manuscript aims to address gaps in understanding factors that influence individual choices in preventative health care particularly lung cancer screening. While providing an interesting insight into experiences for the study cohort, the sample size is relatively small and the findings do not appear specific to lung cancer screening, even though the interview questions focussed on screening. The most important finding of this study appears to be the impact of housing instability. However the data presented do not link this tightly to choice in lung cancer screening. Rather this appears (quite logically) to affect health and attitudes more globally. This study describes of the impact of housing instability and income (as markers of SEC disadvantage) on general attitudes towards health and smoking cessation but is limited in its ability to "unpack" detailed reasons to avoid screening. The profound effect of housing instability appears to override other considerations. As a result more detailed reasons may be hard to elicit.

Other comments:

- the screen selection threshold for PLCOm2012 at 2% is higher than some published data and this is worth acknowledging in the discussion

- not clear whether the "derived rapport" technique was used across all three sites or not - could this introduce bias?

- description of sites would be of value for distribution of demographics in the denominator groups

- there are likely to be factors that influence screening uptake that can't be detected in this small sample (noting the very low levels of uptake detected in the USA - Pham Clin Lung Cancer. 2020 May;21(3):e206-e211. )

- the "judgmental and insensitive" clinical encounters warrant further discussion esp. for future research

While the comments from interviewees were very interesting, the sample is small and the findings - dominated by housing status - are not clearly shown to be linked to rejection of LC screening over and above other preventative health interventions.

Reviewer #3: I found the paper very compelling and thorough - and, distressing. As a strong advocate for both LDCT screening and tobacco cessation, it is housing (and associated economic issues) that is a critical factor - which is yet for another discipline to address.

Two typos: one is the difference between the $30 - 50,000 (line 248) or line 173: $30 - 59,000.

Line 323 should be 'groups' rather than 'group'.

6. PLOS authors have the option to publish the peer review history of their article (what does this mean?). If published, this will include your full peer review and any attached files.

Reviewer #1: No

Reviewer #2: No

Reviewer #3: **Yes: **Carolyn Dresler, MD, MPA

---

## [Author Response · Author response to Decision Letter 0]

1 Apr 2021

Dear Dr. Cummings,

Thank you for your comments and feedback on the manuscript PONE-D-21-01483 Advancing Health Equity in Cancer Care: The Lived Experiences of Poverty and Access to Lung Cancer Screening.

Kindly find attached to this letter revised versions of the manuscript (with tracked changes and an unmarked copy) based on your feedback. Our point-by-point changes/ comments are detailed below:

Thank you we have edited the file to match the formatting requirements of PLOS ONE.

2. In your Methods section, please provide additional information about the participant recruitment method and the demographic details of your participants. Please ensure you have provided sufficient details to replicate the analyses such as:

a) the recruitment date range (month and year),

b) a description of how participants were recruited, and

c) descriptions of where participants were recruited and where the research took place (e.g. name of clincial site).

We have now included more details in the methods section about the recruitment date and specified how we used derived rapport to recruit non-screeners. We have also included a description of the primary care sites through which participants were recruited. To protect the identity of participants we are not disclosing the names of the clinical sites. 

3. Please state in the manuscript Methods:

- Why written consent could not be obtained

- Whether the Institutional Review Board (IRB) approved use of oral consent

- How oral consent was documented

Thank you, we have now included in detail the process of verbal consent and the approval of this process from our IRB. 

'AS is supported by a Postdoctoral Fellowship Award from the Canadian Institutes for Health Research. AL is supported by a New Investigator Award from the Canadian Institutes for Health Research, as Clinician Scientist by the Department of Family Medicine at the University of Toronto and as Chair of Implementation Science at the Peter Gilgan Centre for Women’s Cancers at Women’s College Hospital in partnership with the Canadian Cancer Society. AL is the Provincial Primary Care Lead for Cancer Screening at Ontario Health (Cancer Care Ontario). GL is supported by the Lusi Wong Early Detection of Lung Cancer Program and the Alan B. Brown Chair. GL has received funding from an unrestricted Boehringer Ingerheim grant to develop and implement electronic methods of identifying patients suitable for lung cancer screening in family physician offices. PS is a Clinician Scientist funded by Centre of Addiction and Mental Health, Department of Family and Community Medicine and the Ministry of Health and Long Term Care to direct the STOP program- An Ontario wide smoking cessation program focused on reducing inequities of access for discriminated and underrepresented populations. He also receives support from the Medical Psychiatry Alliance. He has independent grants from the GRAND program, created as an arms-length peer reviewed program for smoking cessation. He is an advisor to both Cancer Care Ontario and Canadian Partnership Against Cancer (CPAC) on tobacco cessation in patients with Cancer. EN is an employee of the CPAC, a pan-Canadian health organization, funded by Health Canada.'

a. Please confirm that this does not alter your adherence to all PLOS ONE policies on sharing data and materials, by including the following statement: "This does not alter our adherence to PLOS ONE policies on sharing data and materials.” (as detailed online in our guide for authors http://journals.plos.org/plosone/s/competing-interests). If there are restrictions on sharing of data and/or materials, please state these.

Please note that we cannot proceed with consideration of your article until this information has been declared.

Thank you for this information. We have now included the statement as recommended in our competing interests. 

Thank you. We have now included this information in our cover letter and appreciate your assistance with the online submission. 

5. Please ensure that you refer to Figure 2 in your text as, if accepted, production will need this reference to link the reader to the figure.

This has been corrected, thank you.

6. Please include your table as part of your main manuscript and remove the individual file. Please note that supplementary tables should remain as separate "supporting information" files.

Thank you, all tables have now been included in the main manuscript file.

7. Please include captions for your Supporting Information files at the end of your manuscript, and update any in-text citations to match accordingly. Please see our Supporting Information guidelines for more information: http://journals.plos.org/plosone/s/supporting-information

Thank you, we have formatted the supporting files with PLOS ONE criteria. 

8. Your ethics statement should only appear in the Methods section of your manuscript. If your ethics statement is written in any section besides the Methods, please delete it from any other section.

Thank you we have placed ethics in the methods section.

Reviewer #1: Sayani and colleagues present the results of a qualitative research study aimed at describing barriers and facilitators to lung cancer screening in low income individuals in Toronto. The methodology and theoretical framework are sound, and the findings are important as countries consider how to target those at the highest risk for lung cancer and are also the most disadvantaged due to socioeconomic status. I have the following comments:

We thank the reviewer for this comment and for highlighting the knowledge gap which our work seeks to address.

1) Overall, the manuscript is quite lengthy of a read at over 5000 words. I understand that PlosOne does not have a limit on words but would encourage the authors to summarize methodology and limitations and strengths sections further.

Thank you for this comment. We have revisited the manuscript and shortened certain areas – in particular the methodology and limitations/ strengths section as suggested. As a qualitative study it is important for us provide thick description, and we have eliminated as much text as possible without drawing away from the process and outcomes of our work. To the best of our knowledge, our manuscript is the first to describe the use of the morphogenetic approach in health services research. Therefore, it is important to fully describe the theory and its applicability to understanding health seeking choices, which we demonstrate through our study for lung cancer screening participation. 

2) Similarly, the authors chose to embed representative quotes into the body of the text which extends the length of the manuscript. I would suggest considering instead constructing a table that pulls from the themes, subthemes and key findings presented in Figure 2 and putting in representative quotes into the table. If the authors chose to leave the quotes embedded, I would suggest reducing the length of the quotes. For example, participant Philip’s quotes could be shortened considerably. In addition, including quotes from the other 18 participants rather than utilizing the same participants might better support the themes and conclusions identified.

Thank you, we have removed all quotations from the paper and added a table instead.

3) In the discussion section, the paragraph that discusses gender-based differences in smoking is not particularly relevant to the focus of the manuscript especially given the low number of female participants and distracts. I would suggest removing.

Thank you for this comment. We have removed this paragraph. 

Reviewer #2: This interesting manuscript aims to address gaps in understanding factors that influence individual choices in preventative health care particularly lung cancer screening. While providing an interesting insight into experiences for the study cohort, the sample size is relatively small and the findings do not appear specific to lung cancer screening, even though the interview questions focussed on screening. The most important finding of this study appears to be the impact of housing instability. However the data presented do not link this tightly to choice in lung cancer screening. Rather this appears (quite logically) to affect health and attitudes more globally. This study describes of the impact of housing instability and income (as markers of SEC disadvantage) on general attitudes towards health and smoking cessation but is limited in its ability to "unpack" detailed reasons to avoid screening. The profound effect of housing instability appears to override other considerations. As a result more detailed reasons may be hard to elicit.

We thank the reviewer for this comment. We have now included a more robust description on the clustering of disadvantage (including low income, housing status and employment etc) in the Results section and how this ties into a trajectory over the life course which is entwined with a higher likelihood of smoking. We have also included a more detailed thematic map (Fig. 2) which ties each of the themes together – specifically housing and screening. Housing is an important determinant of health, however, in the context of smoking behaviour needs special attention as a contributor to smoking patterns as well as barrier to screening. We hope to have addressed this satisfactorily in the manuscript. 

Other comments:

- the screen selection threshold for PLCOm2012 at 2% is higher than some published data and this is worth acknowledging in the discussion

The screen selection threshold of >2% is the screening eligibility criterion being used in the province of Ontario for lung cancer screening. This has now been added to the Methods section. 

- not clear whether the "derived rapport" technique was used across all three sites or not - could this introduce bias?

We have described how derived rapport was used with one health provider – the resulting homogenous nature of participants is described in the limitations of the study. 

- description of sites would be of value for distribution of demographics in the denominator groups

Thank you we have now included a brief description of the clinical sites where participants were recruited from. 

- there are likely to be factors that influence screening uptake that can't be detected in this small sample (noting the very low levels of uptake detected in the USA - Pham Clin Lung Cancer. 2020 May;21(3):e206-e211. )

Thank you for this comment. As a critical qualitative study we sought to contextualise the lived experiences of participants who were eligible for lung cancer screening and their choice to access screening or not. Using this approach we are able to illuminate some policy and training gaps that are specific to increasing the equitable delivery of lung cancer screening.

- the "judgmental and insensitive" clinical encounters warrant further discussion esp. for future research

We appreciate this comment. We have published qualitative interviews with physicians to understand their perspectives on access to lung cancer screening for individuals living with low income. We have now cited this work in the discussion section.

While the comments from interviewees were very interesting, the sample is small and the findings - dominated by housing status - are not clearly shown to be linked to rejection of LC screening over and above other preventative health interventions.

Thank you for this feedback. We reached conceptual saturation with our data through iterative interviewing and peer-debriefing (described in methods). We have tied housing more deeply to the issue of smoking and lung cancer screening in the themes: pathways of disadvantage and the upstream determinants of lung cancer risk and early detection. 

Reviewer #3: I found the paper very compelling and thorough - and, distressing. As a strong advocate for both LDCT screening and tobacco cessation, it is housing (and associated economic issues) that is a critical factor - which is yet for another discipline to address.

Thank you so much for your feedback. We appreciate your comment and hope to move towards a policy discourse which will promote equity in lung cancer screening. 

Two typos: one is the difference between the $30 - 50,000 (line 248) or line 173: $30 - 59,000.

Line 323 should be 'groups' rather than 'group'.

Thank you, these corrections have been made.

We thank the reviewers for their insightful feedback and hope to have addressed the reviewers comments appropriately.

We look forward to your positive consideration,

Yours sincerely,

Ambreen Sayani, MD MSc PhD

---

## [Decision Letter · Decision Letter 1]

23 Apr 2021

Advancing Health Equity in Cancer Care: The Lived Experiences of Poverty and Access to Lung Cancer Screening.

PONE-D-21-01483R1

Dear Dr. Sayani,

We’re pleased to inform you that your manuscript has been judged scientifically suitable for publication and will be formally accepted for publication once it meets all outstanding technical requirements.

Kind regards,

Michael Cummings, PhD

Academic Editor

PLOS ONE

Additional Editor Comments (optional):

Reviewers' comments:

Reviewer's Responses to Questions

**Comments to the Author**

1. If the authors have adequately addressed your comments raised in a previous round of review and you feel that this manuscript is now acceptable for publication, you may indicate that here to bypass the “Comments to the Author” section, enter your conflict of interest statement in the “Confidential to Editor” section, and submit your "Accept" recommendation.

Reviewer #1: All comments have been addressed

Reviewer #2: All comments have been addressed

2. Is the manuscript technically sound, and do the data support the conclusions?

Reviewer #1: Yes

Reviewer #2: Yes

3. Has the statistical analysis been performed appropriately and rigorously? 

Reviewer #1: N/A

Reviewer #2: N/A

4. Have the authors made all data underlying the findings in their manuscript fully available?

Reviewer #1: Yes

Reviewer #2: Yes

5. Is the manuscript presented in an intelligible fashion and written in standard English?

Reviewer #1: Yes

Reviewer #2: Yes

6. Review Comments to the Author

Reviewer #1: Thank you, all of my comments have been addressed. The work presented here is important and the authors should be complemented.

Reviewer #2: Thank you to the authors for this comprehensive revision.  The paper has improved in response to reviews and my comments have been satisfactorily addressed.  The inclusion of quotes from study subjects in Table 2 works well.  The limitations of the study and areas for further research are presented well in the discussion.

7. PLOS authors have the option to publish the peer review history of their article (what does this mean?). If published, this will include your full peer review and any attached files.

Reviewer #1: No

Reviewer #2: No

---

## [Editor Report · Acceptance letter]

27 Apr 2021

PONE-D-21-01483R1 

Advancing health equity in cancer care: the lived experiences of poverty and access to lung cancer screening. 

Dear Dr. Sayani:

I'm pleased to inform you that your manuscript has been deemed suitable for publication in PLOS ONE. Congratulations! Your manuscript is now with our production department. 

Kind regards, 

on behalf of

Dr. Michael Cummings 

Academic Editor

PLOS ONE